evolution

ancient DNA, brown bear, mitochondrial genomes, Pleistocene megafauna

**Authors for correspondence:**
Takahiro Segawa
e-mail: tsegawa@yamanashi.ac.jp
Naoki Kohno
e-mail: kohno@kahaku.go.jp
Hidenori Nishihara
e-mail: hnishiha@bio.titech.ac.jp

†These authors contributed equally.

# Ancient DNA reveals multiple origins and migration waves of extinct Japanese brown bear lineages

Takahiro Segawa[1,†], Takahiro Yonezawa[2,†], Hiroshi Mori[3,†], Ayumi Akiyoshi[4], Morten E. Allentoft[5,6], Ayako Kohno[7], Fuyuki Tokanai[8], Eske Willerslev[6,9,10], Naoki Kohno[7,11] and Hidenori Nishihara[12]

[1]Center for Life Science Research, University of Yamanashi, 1110 Shimokato, Chuo, Yamanashi, Japan
[2]Tokyo University of Agriculture, 1737 Funako, Atsugi City, Kanagawa, Japan
[3]National Institute of Genetics, Yata 1111, Mishima City, Shizuoka, Japan
[4]National Institute of Polar Research, Midori-cho 10-3, Tachikawa City, Tokyo, Japan
[5]Trace and Environmental DNA (TrEnD) Laboratory, School of Molecular and Life Sciences, Curtin University, Bentley, Western Australia 6102, Australia
[6]Lundbeck Foundation GeoGenetics Centre, Globe Institute, University of Copenhagen, Copenhagen, Denmark
[7]Department of Geology and Paleontology, National Museum of Nature and Science, Tokyo, Amakubo, Tsukuba, Ibaraki, Japan
[8]Faculty of Science, Yamagata University, Jonan 4-3-16, Yonezawa City, Yamagata 990-3101, Japan
[9]Department of Zoology, University of Cambridge, Cambridge, UK
[10]Wellcome Trust Sanger Institute, Hinxton, UK
[11]Graduate School of Life and Environmental Sciences, University of Tsukuba, Tennoudai, Tsukuba, Ibaraki, Japan
[12]School of Life Science and Technology, Tokyo Institute of Technology, 4259-S2-17 Nagatsuta-cho, Midori-ku, Yokohama 226-8501, Japan

TS, 0000-0002-3111-708X; NK, 0000-0001-5329-4063; HN, 0000-0002-5843-9994

Little is known about how mammalian biogeography on islands was affected by sea-level fluctuations. In the Japanese Archipelago, brown bears (*Ursus arctos*) currently inhabit only Hokkaido, the northern island, but Pleistocene fossils indicate a past distribution throughout Honshu, Japan's largest island. However, the difficulty of recovering ancient DNA from fossils in temperate East Asia has limited our understanding of their evolutionary history. Here, we analysed mitochondrial DNA from a 32 500-year-old brown bear fossil from Honshu. Our results show that this individual belonged to a previously unknown lineage that split approximately 160 Ka from its sister lineage, the southern Hokkaido clade. This divergence time and

fossil record suggest that brown bears migrated from the Eurasian continent to Honshu at least twice; the first population was an early-diverging lineage (greater than 340 Ka), and the second migrated via Hokkaido after approximately 160 Ka, during the ice age. Thus, glacial-age sea-level falls might have facilitated migrations of large mammals more frequently than previously thought, which may have had a substantial impact on ecosystem dynamics in these isolated islands.

## 1. Introduction

The glacial-interglacial cycles and related sea-level fluctuations during the middle and late Pleistocene are believed to have had a substantial impact on megafauna evolution and population demographics, but details at the molecular level are sparse about how the migration of large mammals onto isolated islands was affected by the dramatic transgression-regression cycles [1]. On the Japanese Archipelago, the opportunities for large mammals to migrate from the Eurasian continent onto the islands have been considered very limited throughout the middle and late Pleistocene, with only a few exceptions when sea level declined drastically during glacial periods (e.g. Yoshikawa *et al.* [2]). For Honshu Island, the largest island of the Japanese Archipelago, various large animals such as the Naumann's elephant (*Palaeoloxodon naumanni*) and the giant deer (*Sinomegaceros yabei*) were present until the late Pleistocene [1,3,4], although most of these species do not currently inhabit Honshu Island. Determining the palaeobiogeographic origins of these extinct large animals is essential for understanding how the unique mammalian fauna became established in the Archipelago and more generally how ecosystems in isolated regions are affected by colonization and extinction of the large animals.

Brown bears (*Ursus arctos*), one of the largest extant carnivores, occupy an important niche in coniferous forests of the Northern Hemisphere. On the Japanese Archipelago, brown bears currently inhabit only Hokkaido Island, the northern island of Japan [5] (figure 1). The brown bear population on Hokkaido Island comprises three maternal lineages, namely the central, eastern and southern Hokkaido lineages [9]. In particular, the southern Hokkaido lineage is closely related to North American populations, and the central Hokkaido lineage is closely related to eastern European populations, indicating that the three lineages independently migrated to Hokkaido Island at different times [6]. Based on an analysis of the mitochondrial genome, it is estimated that the southern, eastern and central Hokkaido populations migrated during periods spanning 194–36 Ka, 165–42 Ka and 53–27 Ka, respectively [6]. Unfortunately, few fossils of large mammals except the brown bear from the middle to late Pleistocene have been found on Hokkaido Island because of the lack of suitable sediments, and therefore palaeontological information is lacking concerning the timing of brown bear migration to Hokkaido Island and how long they remained there.

By contrast, brown bears were widespread in Honshu during the middle and late Pleistocene evidenced by a large number of brown bear fossils dating from 340 to 20 Ka [7]. For reasons unknown, the bears were extirpated by the end of the late Pleistocene in Honshu. Knowledge of the evolutionary history of these brown bears is crucial for understanding the occurrence and extinction of megafauna in the Japanese Archipelago (e.g. Kamei *et al.* [4]) because the bear is considered to be at the top of the food chain on the Archipelago. However, it is currently uncertain which brown bear lineage the Honshu population is phylogenetically related to, and it is also not known when and by what route they migrated to the Archipelago. The oldest known fossil of the Honshu brown bear is 340 Ka [10], and therefore brown bears in Honshu are considered to have migrated during the Chibanian age (middle Pleistocene) [1]. Curiously, this age is much older than the estimated time for migration of the three aforementioned Hokkaido populations [6], making it important to clarify the genetic relationship between these populations.

There are two possible routes for large terrestrial mammals migrating from the Eurasian continent to Honshu Island: the Sakhalin-Hokkaido route and the Korean route [11–13] (figure 1). The extant three lineages of the Hokkaido brown bears are considered to have migrated across a land bridge between Sakhalin and Hokkaido Islands, as that bridge formed intermittently during glacial periods of the Pleistocene [1,4]. By contrast, large mammals are believed to have rarely migrated between Hokkaido and Honshu across the Tsugaru Strait [14], which poses a formidable barrier to animal migration between these islands; this barrier is called the Blakiston Line. Importantly, the present-day fauna differs greatly between the two islands; the fauna on Hokkaido is related to northern Asian fauna, whereas that on Honshu is related to animals from eastern Asia [15]. Regardless of the route, the opportunities for large-animal migrations to Honshu Island were limited throughout the Pleistocene.

DNA sequence data for the Pleistocene brown bears of Honshu Island could resolve when and by which route the brown bear population arrived to the Island. However, the preservation of DNA in

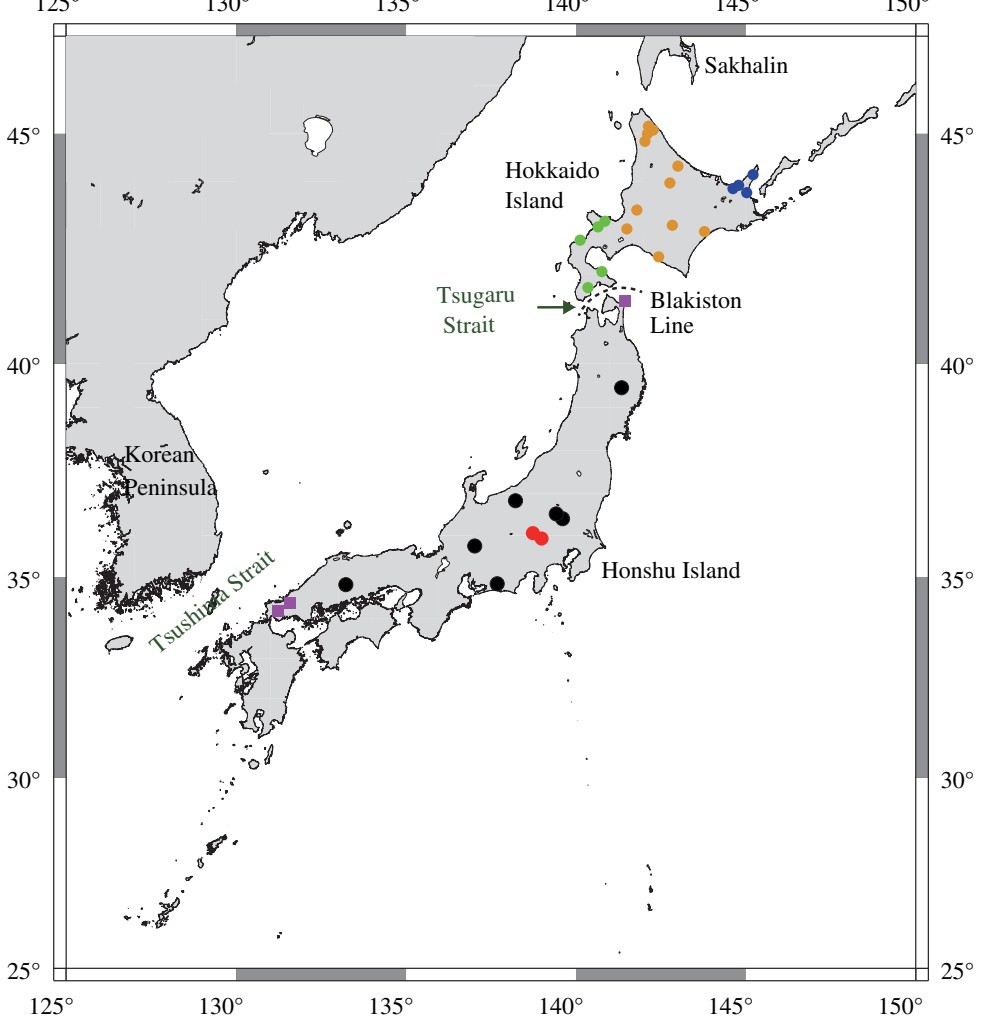

**Figure 1.** Distribution of extant brown bears in Hokkaido and fossil localities in Honshu [6–8]. Symbols represent the various ages of fossils in Honshu as follows: black circle, late Pleistocene; purple square, Chibanian (middle Pleistocene); red circle, late Pleistocene (this study). Extant brown bears comprise three populations inhabiting southern (green), central (yellow) and eastern (blue) Hokkaido. The map was created with GMT 5 software (https://www.soest.hawaii.edu/gmt).

fossils from middle latitudes such as Japan is generally very poor because of the hot and humid climate as well as the acidic soil, which accelerates DNA degradation [16]. Indeed, analyses of ancient DNA from Japanese sources is lagging relative to efforts in other countries [17], and no ancient DNA has been reported from Pleistocene fossils of brown bears from Japan or East Asia. In this study, we carried out an analysis of ancient DNA obtained from a Pleistocene brown bear fossil from Honshu Island to elucidate the process by which brown bears became distributed in the Japanese Archipelago.

## 2. Material and methods

### 2.1. Sample descriptions, radiocarbon dating and stable isotope measurements

The samples for the ancient DNA analysis described here comprised a petrous and a radius (JBB-32 K) and a canine tooth (JBB-19 K) collected from different localities (electronic supplementary material, figure S1). Only one skull with petrosals of definitive brown bear is known from Honshu Island [18]. The specimen, i.e. JBB-32 K, was excavated in the 1960s at Oinuana cave in Ueno Village, Tano County, Gunma Prefecture, Japan (figure 1). This specimen has been stored at the Ueno Village Board of Education, Gunma, Japan. Miyazaki *et al.* [18] fully described and identified the specimen as a brown bear, consisting of a braincase with both left and right petrosals, a left mandibular fragment, and some postcranial bones including the radius, humerus, tibia and vertebrae belonging to a single

individual. The canine of this individual was relatively small, but its root was completely closed based on its alveolus, and therefore the individual was considered to be an old adult female [18]. According to Miyazaki *et al.* [18], the size of this individual reaches the average size of present-day male brown bears distributed in Hokkaido. It is interesting to note that the body size reduction of large mammals in islands are thought to be a general tendency, which is well known as 'island rule' [19]. In this regard, if the phylogenetic relationship of this individual to other populations can be clarified, it can also be discussed whether such an inclination is to be, though it is not the focus of this study.

The canine specimen was collected as an isolated tooth in May 1956 from a small cave located in Chichibu City, Saitama Prefecture, Japan. Its crown was very large and identical to the size and proportion of those of brown bears [20], and its root was completely filled in by dentin and surrounded by relatively thick cementum, suggesting that the individual was an old adult male of the brown bear (electronic supplementary material, figure S1). The canine's dimensions were: mesio-distal diameter, 19.30 mm; bucco-lingual diameter, 14.66 mm; crown height, 21.93 mm; root depth, 26.87 mm; root thickness, 20.16 mm; root length, 53.99 mm. This tooth has been stored at the Saitama Museum of Natural History, Nagatoro, Saitama, Japan, and it bears the registration number SMNH-VeF 226.

The geological ages, in terms of before present (BP), for JBB-32 K and JBB-19 K were determined by $^{14}$C analysis. Pretreatment for the samples before graphitization was conducted at the National Museum of Nature and Science, Japan. A 500 mg sample of bone powder and tooth dentin were extracted from the radius of the same individual as the braincase (JBB-32 K) and the root of the canine (JBB-19 K) using an Emax evolution drill (Nakanishi, Japan). The bone powder was cleaned with ultrasound, and the alkali-soluble organic substances were removed with 0.1 N NaOH for 10 min. The inorganic matter such as the hydroxyapatite was also dissolved with 1.2 N HCl. The extract was separated into a supernatant and a residue by a centrifuge, and finally freeze-dried for 24 h. The remaining organic matter was heated at 95° C for 12 h in pure water. After that, the centrifuged supernatant was suction-filtered and collected in a vial and freeze-dried to obtain the gelatin collagen. Gelatin collagen was graphitized by an iron-catalysed hydrogen reduction method using a graphite preparation system consisting of an elemental analyser and a glass vacuum line. Radiocarbon dating was performed with a compact accelerator mass spectrometry system based on a 0.5-MV Pelletron accelerator developed by National Electrostatics Corporation at Yamagata University Accelerator Mass Spectrometry (YU-AMS), Japan [21]. The radiocarbon dates were converted to calendar ages (cal. BP) using the program OxCal v. 4.2 with a calibration curve based on IntCal13 [22]. Carbon and nitrogen stable isotope ratios were measured from the collagen sample using an elemental analyser coupled with an isotope ratio mass spectrometer, also at YU-AMS. Stable isotope compositions are reported using the $\delta$ notation (i.e. $\delta$13C and $\delta$15N).

## 2.2. Extraction of ancient DNA from fossils

All work with the ancient DNA, before genome library amplification, was conducted in a dedicated clean room for ancient DNA at the National Institute of Polar Research (NIPR) for JBB-19 K and JBB-32 K, Tokyo, Japan, and at the Centre for GeoGenetics, Copenhagen, Denmark for JBB-19 K. We extracted 100−300 mg of petrosal bones and a radial bone (JBB-32 K) and canine (JBB-19 K) per DNA extraction session. We followed the DNA extraction procedure based on silica pellets in solution as described in Orlando *et al.* [23] for JBB-19 K and JBB-32 K. We performed a pre-digestion by incubating the sample in 1 ml digestion buffer (0.5 M EDTA, 0.2 mg ml$^{-1}$ proteinase K, 0.5% *N*-lauryl sarcosyl) [24] at 42°C for 10 min. After centrifugation for 2 min at 2000$g$, the supernatant (the pre-digest) was removed, and the pellet was incubated for 36 h at 42°C in 3 ml fresh digestion buffer. After centrifugation for 2 min at 2000$g$, the supernatant was incubated with 100 µl silica beads and 40 ml binding buffer (Qiagen PB buffer; 25 mM NaCl, 87 mM sodium acetate) for 3 h at room temperature. The supernatant was discarded, and the pellet was washed twice with 1 ml of 80% ethanol before eluting the DNA with 100 µl elution buffer from the Qiagen kit.

We also performed DNA extraction from sample JBB-19 K with a silica column-based approach as described in [25]. We pre-digested the sample in 1 ml extraction buffer (0.45 M EDTA, 0.25 mg ml$^{-1}$ proteinase K) for 10 min at 37°C. After centrifugation for 2 min at 2000$g$, we discarded the supernatant and incubated the pellet for 36 h at 37°C in 2 ml of fresh extraction buffer. After centrifugation for 2 min at 10 000$g$, we transferred the supernatant mixed with 13 ml binding buffer (Qiagen PB buffer; 90 mM NaCl, 12.5 mM sodium acetate) [26] on a Zymo-Spin V reservoir (Zymo Research) fitted on a MinElute column (Qiagen). The reservoir-MinElute device was centrifuged at 270$g$ for 10 min. The column was washed twice with 750 µl PE buffer (Qiagen) and pelleted at 10 000$g$ for 1 min. The DNA was eluted

twice with 50 µl EB buffer (Qiagen) after a 10 min incubation at 37°C. The same procedure without tooth powder was conducted as a negative control to identify any contaminating DNA.

We removed the most frequent post-mortem DNA damage (cytosines deaminated to yield uracils [26]) from two (experiment 7 and experiment 8 in the electronic supplementary material, table S1) of the 11 experiments of extracted DNA, after incubation with USER Enzyme (New England BioLabs) and NEBNext FFPE DNA Repair Mix (New England BioLabs). USER Enzyme is a mixture of uracil glycosylase and the DNA glycosylase-lyase endonuclease VIII, and NEBNext FFPE DNA Repair Mix is a cocktail of enzymes formulated to repair DNA and is specifically optimized and validated for the repair of FFPE DNA samples. For the USER Enzyme reaction, we incubated 16.5 µl DNA extract with 5 µl USER Enzyme mix for 3 h at 37°C. The mixture was purified using the MinElute PCR Purification kit (Qiagen) and eluted with 20 µl EB buffer at 37°C for 10 min. For FFPE DNA Repair Mix, we incubated 22.3 µl DNA extract with 2.7 µl FFPE DNA Repair Buffer and 0.8 µl NEBNext FFPE DNA Repair Mix for 1 h at 20°C. The repaired DNA solution was immediately used to generate a sequencing library. For the remaining nine experiments, the genome libraries were generated without removing the most frequent post-mortem DNA damage.

## 2.3. Genome library construction and illumina sequencing

Double-stranded DNA libraries were prepared using an NEBNext Quick DNA Library Prep Master Mix Set for 454 (New England BioLabs) at the Centre for GeoGenetics, Denmark and NIPR. Libraries were prepared with 20 µl DNA extract or damage-repaired DNA extract without DNA fragmentation. Illumina sequencing adapters were added to the end-repair reaction at a final concentration of 0.25 µM together with 1 U Quick T4 DNA ligase. DNA fragments in each reaction were purified using the MinElute PCR Purification kit and were eluted with 20 µl EB buffer at 37°C for 10 min. A polymerase chain reaction (PCR) amplification reaction of each genomic library was prepared in a laboratory designed specifically for work with ancient DNA and then transported to PCR thermal cyclers in a conventional laboratory, which was physically separated. Each 50 µl reaction contained KAPA HiFi HotStart Uracil + ReadyMix (Kapa Biosystems) or AmpliTaq Gold DNA polymerase (Life Technologies). Each genomic library was amplified for 12−14 cycles and was then cleaned with the MinElute PCR Purification kit.

The PCR products were separated by 3% agarose gel electrophoresis and purified using a NucleoSpin Gel and PCR Clean-up kit (Macherey-Nagel). The quality of the library and the library size were assessed with an Agilent 2100 Bioanalyzer using the High Sensitivity DNA kit (Agilent). Sequencing was carried out on an Illumina MiSeq at the NIPR, an Illumina HiSeq 4000 at the Danish National High-Throughput DNA Sequencing Centre, University of Copenhagen, Denmark, and an Illumina HiSeq XTen at the Beijing Genomics Institute, China. Together, this generated 1 158 928 336 sequencing reads. The sequencing information is listed in the electronic supplementary material, table S1.

## 2.4. Bioinformatics

We discarded the single-end reads that were mapped to the PhiX genome sequence using BOWTIE 2 v. 2.2.3 [27] with default parameters. We then removed the adapter sequences in the reads and conducted quality filtering using fastp v. 0.12.5 with default parameters [28]. The high-quality reads derived from the Honshu brown bear mitochondrial genome were identified as follows: (i) all of the high-quality reads were subjected to mapping using BWA-MEM v. 0.7.17 with default parameters [29] against the *U. arctos* mitochondrial genome sequence (NC_003427.1); (ii) the PCR duplicate reads were removed using the rmdup function of seqkit v. 0.15 [30]; and (iii) a consensus sequence of the mitochondrial genome of the Honshu brown bear was reconstructed using INTEGRATIVE GENOMICS VIEWER [31] and with subsequent manual curation with read coverage greater than 1 (we manually replaced the nucleotides to $N$ in the site that the read coverage $\leq 0$). The nucleotide misincorporation rate for the Honshu brown bear was calculated using the mitochondrial genome sequence as the reference for MAPDAMAGE2 [32] (electronic supplementary material, figure S2).

## 2.5. Phylogenetic analysis based on the complete mitochondrial DNA sequence data

We retrieved 95 published near-complete (greater than 16 kbp) mitochondrial DNA (mtDNA) sequences of brown bears, including all known individuals from clade 3a2 (central Hokkaido), 3b (eastern Hokkaido, Etorofu and Kunashiri) and 4 (southern Hokkaido and North America) as well as

representative individuals from all other clades. The concatenated sequence data, consisting of the 13 coding sequences, 22 transfer RNAs (tRNAs) and two ribosomal RNAs (rRNAs) but excluding overlapping regions, as well as the D-loop excluding a hypervariable region, were aligned with MAFFT [33], and misaligned segments of the alignment were then corrected. Sequence data for 16 084 sites were separated into 10 partitions according to the three codon positions and the mtDNA coding strand, as well as tRNAs (H- and L-strand), rRNAs and the D-loop region, in accordance with the PARTITIONFINDER2 [34]. The maximum-likelihood (ML) tree was inferred using IQ-TREE v. 1.6.10 [35] with the GTR + I+$\Gamma$ model, and 1000 standard bootstrap replicates were used to estimate nodal support.

## 2.6. Estimating coalescence times among *Ursus arctos*

The tip-dating analysis was conducted for estimating the coalescence times among *U. arctos* by Bayesian inference with BEAST v. 1.10.4 [36]. The alignment used for inferring the ML tree, as mentioned above, was used for this analysis. The HKY + I+$\Gamma$ model was used for the nucleotide substitution model. The $^{14}$C dates converted to calendar ages were used for the tip-dating analysis. Because the Siberian Pleistocene individual (MH255807, Rey-Iglesia *et al.* [37]) and French Pleistocene individual (EU497665, Bon *et al.* [38]) lack accurate $^{14}$C dating information, the dates of their tip nodes were estimated by Bayesian inference under a wide range of the uniform prior distributions (50 000 ± 50 000 BP). Because every tip date should have prior distributions in this setting of analysis, we arbitrarily provided the uniform prior distributions of 0 ± 10 BP for modern samples, and $^{14}$C dating ± 100 BP for ancient samples reported by previous works. The uniform prior distributions of 32 719–32 209 BP was given for our sample (JBB-32 K: see below). The relaxed clock method with uncorrelated lognormal distribution of the molecular evolutionary rate was applied under the assumption of a coalescent process with constant population size. A Markov chain Monte Carlo experiment was conducted with the length of 100 000 000 generations, and trees were sampled per 10 000 generations. The convergence of each parameter was confirmed by checking that every effective sample size was greater than 200 with TRACER v. 1.7 (http://tree.bio.ed.ac.uk/software/tracer/).

Taking into account the different effect of time dependency of the molecular evolutionary rates among genes [39], coalescence times were independently estimated with three datasets. The first dataset consisted of whole mitochondrial genomes with 10 partitions used for the inference of the aforementioned ML tree. The total length of this dataset was 16 084 bp. The second dataset consisted of 13 entire protein-coding genes with six partitions: the first, the second and the third positions of concatenated 12 protein-coding genes in the H-strand and the ND6. The total length of this dataset was 11 214 bp. The third dataset consisted of the third codon positions of the 13 protein-coding genes with two partitions: the third positions of the concatenated 12 protein-coding genes and the ND6. The total length of this dataset was 3738 bp.

## 2.7. Estimating ancestral geographical distributions of *Ursus arctos*

The geographical distributions of *U. arctos* were estimated with the program BAYESTRAITS v. 3.0.2 (http://www.evolution.rdg.ac.uk/BayesTraitsV3.0.2/BayesTraitsV3.0.2.html). A geographical model [40] of the Brownian motion on a three-dimensional Cartesian coordinates system was used to map the geographical coordinates (longitude and latitude) of the ancestral nodes with the Markov chain Monte Carlo of 1 000 000 generations. As for information pertaining to genealogy and time intervals between nodes, the coalescence time tree based on mitochondrial genome data, as mentioned above, was applied for this analysis. The ancestral geographical coordinates were sampled per 1000 generations. The first 200 000 generations were discarded as burn-in. TRACER v. 1.7 was used to confirm the convergence of parameters and estimate the summary statistics.

## 2.8. Phylogeographic analysis based on the partial mitochondrial DNA sequence data

All available partial mtDNA sequence data (*cytb* and D-loop) were retrieved from NCBI, and accession numbers for these sequences are shown in the electronic supplementary material, table S3. The nucleotide sequences of *cytb* and D-loop were aligned with the MAFFT program together with the mitochondrial genomes, respectively.

ML trees were inferred by the IQ-TREE v. 2.0.4 [41]. The nucleotide substitution models were selected based on the Bayesian information criterion using MODELFINDER [42]. The HKY + F+I+$\Gamma$ model was selected for the *cytb* supermatrix, and the K3Pu + F+I+$\Gamma$ model was selected for the D-loop supermatrix.

To evaluate the confidence of the internal branches, ultrafast bootstrap [43] was applied with 1000 replicates.

# 3. Results

## 3.1. Radiocarbon measurements and isotopic signatures

The geochronological age of the extinct Honshu brown bear specimens was estimated based on a $^{14}$C analysis. For the JBB-32 K sample, the estimated age was 28 459 ± 86 BP, with the estimated calendar-year age range being 32 719–32 209 cal. BP; for the JBB-19 K sample, the estimated age was 16 010 ± 60 BP, with the calendar-year range being 19 434–19 212 cal. BP. These ages correspond to the late Pleistocene [44] (electronic supplementary material, table S2). The stable isotope compositions ($\delta^{13}$C and $\delta^{15}$N) of JBB-32 K and JBB-19 K were measured to be −17.45‰ and 8.68‰ and −17.58‰ and 10.11‰, respectively (electronic supplementary material, figure S3). These isotopic values differed from those of current Eurasian and North American brown bears (omnivorous mammals that eat medium- to large-sized mammals and plant material including fruits) yet were very similar to those of the extinct North American short-faced bear (*Arctodus simus*) that has been considered to be more carnivorous, based on the relatively high concentration of $\delta^{15}$N [37,45], suggesting that the Honshu brown bears were also highly carnivorous.

## 3.2. Sequencing of mitochondrial DNA from samples of the Pleistocene Honshu brown bear

We extracted DNA from fossils JBB-32 K and JBB-19 K. The extracted DNA was subjected to 11 independent analyses with next-generation sequencing that yielded four reads (corresponding to two uniquely mapped reads after the PCR duplicate removal) and 4788 reads (corresponding to 1310 uniquely mapped reads after the PCR duplicate removal, comprised 1280 reads from the petrous and 30 reads from the radius) of mtDNA identified among 723 764 255 quality filtering reads from JBB-19 K and 435 164 081 quality filtering reads from JBB-32 K, respectively (electronic supplementary material, table S1). The proportion of endogenous mtDNA (i.e. of all DNA extracted) in these samples was quite low ($5.5 \times 10^{-7}$% for JBB-19 K, $1.1 \times 10^{-3}$% for JBB-32 K), while it typically ranges from $6 \times 10^{-4}$% to $1 \times 10^{-2}$% for Pleistocene fossils depending on the degree of sample preservation [37]. The numbers of endogenous reads depend on the preserved portion and the condition of specimen regardless of geological ages. The petrosal bone is richer in organic matter (25%) than dentin of canine teeth (20%), and the petrous is denser than canine dentin. JBB-32 K was found in the deposit of a horizontal cave and was relatively free from water, while JBB-19 K was collected from the deposit of a vertical cave and was constantly affected by submersion. This must also be the reason why the condition and performance of endogenous reads between the two was different. The reconstructed mtDNA sequence obtained for JBB-32 K covered an 84.1% region of the complete mitochondrial reference sequence (North American brown bear: NC_003427.1) with a 4× average coverage. The two unique mtDNA sequences obtained from JBB-19 K corresponded to nucleotide positions 668–695 (28 bp) and 1232–1262 (31 bp) in the genes *ND2* and *ND4*, respectively. The sequences were identical to those of JBB-32 K as well as most of the other extant brown bears residing in Eurasia. Hereafter, we used the JBB-32 K mtDNA data for evolutionary analysis.

Previous studies suggested that the extant brown bears can be categorized into nine mtDNA clades, namely clades 1, 2a, 2b (polar bear), 3a1, 3a2, 3b, 4, 5, and the early-diverging Himalayan clade [37,46–49]. We constructed an ML tree which revealed that the Honshu brown bear (JBB-32 K) belongs to clade 4 but constituting a previously unknown lineage sitting as a sister group to the southern Hokkaido brown bears; furthermore, a North American brown bear is a sister group to these lineages within clade 4. These relationships were supported by 100% bootstrap values (electronic supplementary material, figure S4).

## 3.3. Coalescence times in the *Ursus arctos* species complex

Figure 2 presents a coalescence time tree for the *U. arctos* species complex based on the complete mitochondrial genome with the relaxed clock model by BEAST. The time of divergence from the most recent common ancestor (tMRCA) of the eastern clade, or the divergence time between Tibetan brown bears (clade 5) and the remaining eastern clade (clade 3 + clade 4), was estimated to be 288 Ka (95% highest posterior density (HPD): 216–367 Ka). The tMRCA of the remaining eastern clade (node A in

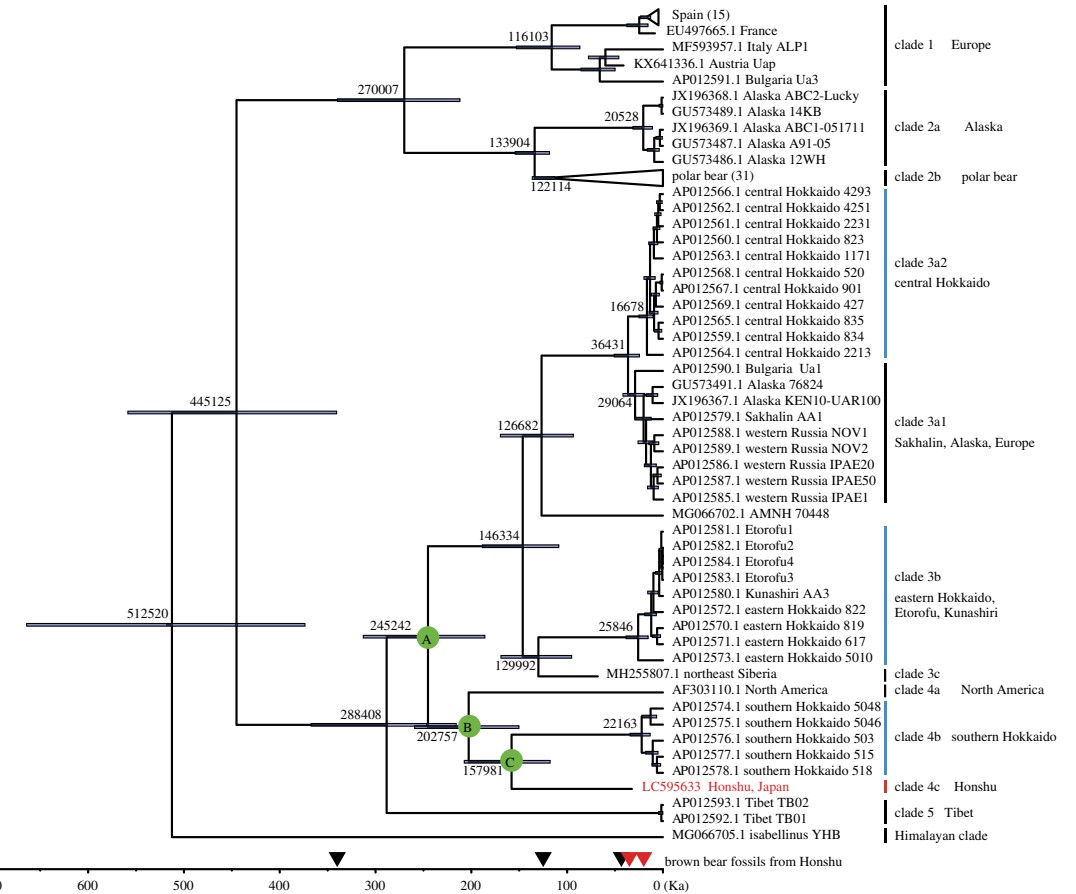

**Figure 2.** Tip-calibrated coalescence times based on the brown bear mtDNA analysed by BEAST v. 1.10.4. Branch labels indicate the median estimated divergence time, and blue bars indicate the 95% highest posterior densities. Coalescence times based on the complete mitochondrial genomes consisting of 13 protein-coding genes, two rRNAs, 22 tRNAs and D-loop. The green circles represent the MRCA for which the geographical distribution was inferred. Clades shown by light blue and red bars include the extant and extinct populations in Hokkaido and Honshu, respectively. The clade with the blue bars includes extant populations in Hokkaido, and the red bar indicates the extinct population in Honshu. Geological ages of brown bear fossils in Honshu that have been estimated previously and in this study are shown in black and red arrowheads on the age axis, respectively [7,10,50].

figure 2), or the split time of clade 3 + clade 4, was estimated as 245 Ka (95% HPD: 186–313 Ka). The tMRCA of clade 4 (node B in figure 2), or the split time between the Honshu-southern Hokkaido group and its closest North American lineage, was estimated as 203 Ka (95% HPD: 150–259 Ka). The split time of the Honshu and the southern Hokkaido lineages (node C in figure 2) was estimated as 158 Ka (95% HPD: 118–207 Ka), and that was, notably, earlier than the tMRCA for each of the three Hokkaido clades. The tMRCA of clade 3 was estimated as 146 Ka (95% HPD: 109–188 Ka). The split time between the eastern Hokkaido lineage and its closest (possibly extinct) Russian lineage was estimated as 130 Ka (95% HPD: 95–169 Ka). The split time between the central Hokkaido lineage and its closest east Europe + Sakhalin + Alaska lineage was estimated as 36 Ka (95% HPD: 25–51 Ka). The coalescence time trees based on the complete sequences for the 13 mtDNA protein-coding genes as well as the third position in each codon of these genes are shown in the electronic supplementary material, figures S5 and S6, respectively. The estimated coalescence times based on these data were essentially consistent with the estimates based on the complete mitochondrial genomes.

## 3.4. Ancestral geographical distribution of the *Ursus arctos* species complex

To investigate the potential migration routes of the extinct Honshu and extant southern Hokkaido brown bears, we retrieved all available partial mtDNA sequences (cytochrome *b* (*cytb*) and D-loop) of brown bears and inferred the phylogenetic trees to identify the closest relatives of the Honshu and the southern Hokkaido clade in the Eurasian continent. None of the modern or ancient brown bears reported in previous studies was found to be the closest relative to the Honshu and the southern

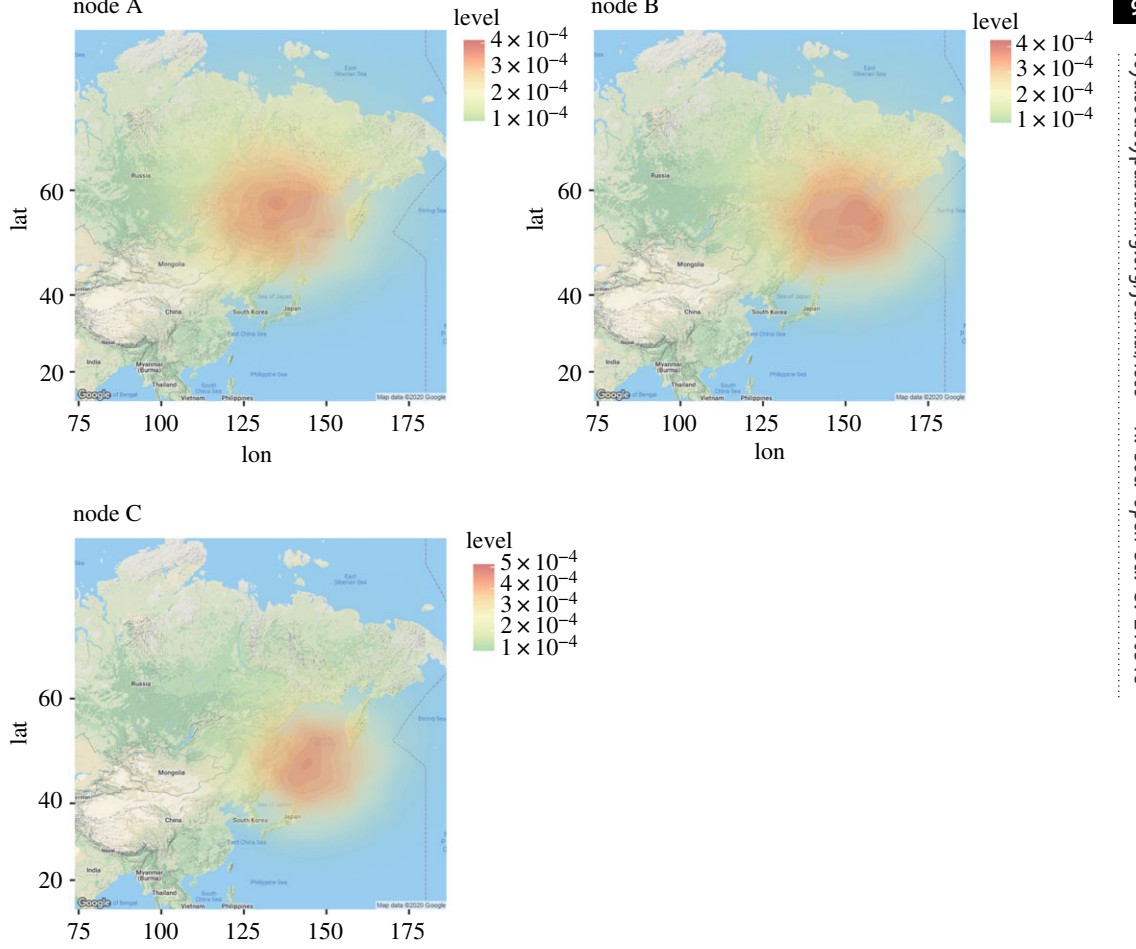

**Figure 3.** Geographical ranges of the ancestral nodes of the brown bears based on Bayesian inference with the geographical model. Posterior distributions based on kernel density estimations are shown. (*a–c*) Geographical range for each of node A, node B and node C. The nodal numbers correspond with those shown in figure 2.

Hokkaido clade based on *cytb* sequence data (electronic supplementary material, figure S7). The D-loop analysis also revealed that no individuals from the Eurasian continent belong to clade 4 (data not shown). Accordingly, their potential migration routes could not be elucidated by this approach but it is obvious that the Honshu brown bear is a previously unknown lineage and does not belong to any known extant bear populations worldwide.

We directly estimated the geographical distribution of each of the ancestral nodes by the Bayesian inference. Geographical coordinates of ancestral nodes were inferred by the geographical model under the coalescent time tree based on the complete mitochondrial genome. The geographical distribution of node A (figure 2), which is the MRCA of clade 3 and clade 4, was estimated to be the area centred on Khabarovsk Krai in the Russian far east (the posterior mean of the longitude and latitude is 137.8 E and 57.4 N) (figure 3). The geographical distribution of node B, which is the MRCA of clade 4 (Honshu, southern Hokkaido and North America), was estimated to be the coastal area encompassing the Sea of Okhotsk (posterior mean, 150.0 E, 56.2 N). The geographical distribution of node C, which is the MRCA of the Honshu and southern Hokkaido populations, was estimated to be the area centred on Sakhalin Island (posterior mean, 144.8 E, 49.3 N).

# 4. Discussion

## 4.1. Multiple migrations of brown bear populations into Honshu Island

This is, to our knowledge, the first successful analysis of ancient DNA from Pleistocene brown bears in the Japanese Archipelago. Our analyses enabled us to solve the phylogenetic origin of the Honshu brown

bear which has been a long-standing debate since the first fossil was reported in 1949 [51]. Here, we elucidated that the Honshu population represents a previously unknown lineage and was the closest relative of the extant southern Hokkaido population, which diverged 158 Ka. This divergence time is at least equivalent to or earlier than the origins of the major clades such as clades 2a, 2b, 3a1, 3a2 and 3b (figure 2), and the three groups within clade 4 are from geographically distant areas, suggesting genealogically more distant relationships. Therefore, we propose that the traditional clade 4 can be separated into the following three clades: clade 4a (North America), clade 4b (southern Hokkaido) and clade 4c (Honshu, extinct). Although the three maternal lineages are geographically distant, the clade 4 brown bears might have inhabited a broad region encompassing Eurasia, and North America during the Pleistocene, as also suggested by the partial mtDNA sequence of clade 4 from a Pleistocene brown bear fossil in Spain [52]. The clade 4 brown bears were later displaced by the clade 3 populations that currently predominate in this vast intercontinental region. Indeed, the Holocene brown bears on Sakhalin Island (753–9620 years BP) belonged to clade 3, the same as the eastern and central Hokkaido lineages [53], which is consistent with our hypothesis.

By contrast, the oldest known record of the Honshu brown bear is reported to be approximately 340 Ka [10]. This fossil is known from a limestone quarry at the Shimokita Peninsula in Aomori Prefecture, northernmost of Honshu Island. The fossil bearing bed is composed of marine sand that covers a tidal notch formed on the limestone at about 80 m above sea level. The well-established correlation between sea levels of coastal terraces, tephrochronologies for some tephras covered on terraces, and geological ages based on oxygen isotopes in this area suggested that the fossil bearing sediment was deposited during the Marine Isotopic Stage 9 (300–337 Ka), one of the warm periods in the late middle Pleistocene [54]. Based on this palaeontological evidence, brown bears have been regarded as one of the 'early colonists' among mammals on Honshu Island, having migrated to the island during the Chibanian (middle Pleistocene) [1]. It should be noted that this oldest fossil significantly predates the emergence of the late Pleistocene Honshu lineage analysed in this study (158 Ka; 95% HPD 118–207 Ka) and is even older than the tMRCA of clade 4 (Honshu + southern Hokkaido + North American clades) estimated as 203 Ka (95% HPD: 150–259 Ka) (figure 2). Because the clade 4a brown bears are distributed in North America [48], it is likely that the MRCA of clade 4 did not inhabit the Japanese Archipelago. This finding indicates that the Honshu brown bear during the Chibanian, as an 'early colonist', is genetically distinct from bears of the late Pleistocene at least in terms of maternal lineages, suggesting that there were multiple, independent migration events of brown bears from the Eurasian continent to Honshu Island during the middle and late Pleistocene. This is a very surprising result because it has generally been believed that the opportunities for migration of forest-dwelling, large animals from Eurasia to Honshu were very limited throughout the Pleistocene because of fewer land-bridge connections between the continent and Honshu Island, especially after the middle Pleistocene [14]. This result suggests that large mammals may have migrated from the continent to Honshu more frequently than previously thought.

## 4.2. Migration times and routes of Honshu brown bear populations

There were two possible routes for brown bear migration from the Eurasian continent to Honshu Island: the Sakhalin-Hokkaido route and the Korean route [11–13]. The land bridge was formed between the Korean Peninsula and Honshu because of low sea levels *ca*. 430 Ka during the marine isotope stage (MIS) 12 (478–424 Ka) and *ca*. 630 Ka during MIS 16 (676–621 Ka) [2,55], and large mammals such as Naumann's elephants are considered to have migrated via this route during the former period for instance [55,56]. However, because the oldest known fossil (approx. 340 Ka) of the Honshu brown bear is only the record from that age in the Japanese Islands including Hokkaido, the migration route and/or direction for the first colonization of the Honshu brown bear is presently uncertain despite the excavation site being located just beside the Tsugaru Strait. As for the second ancestral population represented by JBB-32 K (clade 4c), the Sakhalin-Hokkaido route is supported by our results that (i) the southern Hokkaido group is the closest relative of the Honshu brown bear (figure 2); (ii) all known individuals in east Eurasia, including the Maritime Territory of Russia, are not close to the southern Hokkaido or Honshu lineages (electronic supplementary material, figure S7); and (iii) the MRCA of the Honshu and southern Hokkaido clade was estimated to inhabit the area around Sakhalin based on Bayesian inference (figure 3). More importantly, the split time between the Honshu and southern Hokkaido lineages was estimated as approximately 160 Ka (figure 2). In general, the Tsugaru Strait is known to be an important faunal boundary between Hokkaido and Honshu Islands (the Blakiston Line), but a few cases of large mammals migrating beyond this line during the

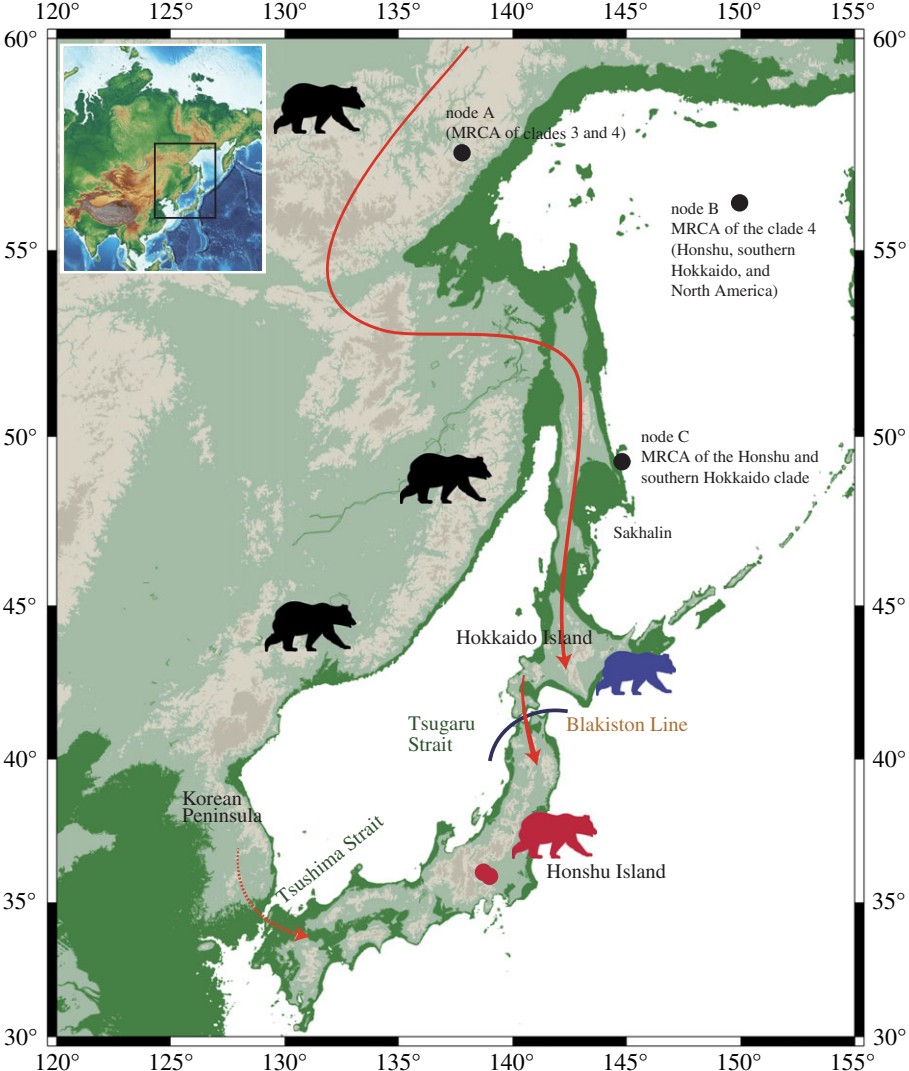

**Figure 4.** Proposed migration routes of the Honshu brown bears. A population of clade 4c migrated into Hokkaido via Sakhalin before the late Pleistocene (blue bear), and a part of that population (red bear) migrated further southwards across the Tsugaru Strait into Honshu approximately 140 Ka (solid red arrows). Alternatively, it is possible that brown bears migrated via the Korean Peninsula (dashed red arrow). The black-bear figures denote ancestral bear populations of extant brown bears in the continent, red circles in Honshu denote locations of the two fossil specimens used in this study, and black circles denote posterior means of the ancestral habitat regions inferred by BAYESTRAITS. Based on the data of Rohling *et al.* [58] and Bintanja *et al.* [59], the areas in green indicate that the sea level dropped by 120 m because of the accumulation of continental ice. The map was drawn using GMT v. 5 along with topographical data from the ETOPO1 Global Relief Model (https://www.ngdc. noaa.gov/mgg/global/).

Pleistocene have been reported. One of the exceptional cases is the migration of Naumann's elephants and giant deer, for which the fossil record supports that they crossed from Honshu to Hokkaido approximately 140 Ka during the glacial period of low sea levels (MIS 6a–6b) [57]. This age is, remarkably, very close to the divergence time between the Honshu and the southern Hokkaido lineages (figure 2; electronic supplementary material, figures S5, S6). Therefore, it is more likely that the brown bear population expanded southwards (from Hokkaido to Honshu) during this glacial period by crossing the Blakiston Line (figure 4). The existence of brown bear fossils of approximately 125 Ka in northernmost Honshu also supports this hypothesis [10]. Such a biotic interchange event in this period involving both large herbivores and carnivores might have greatly affected the Pleistocene fauna across the largest islands of the Japanese Archipelago. Hence, we propose that migration of brown bears into Honshu occurred at least twice; the first brown bear population migrated during MIS 12 or 16 in the Chibanian, and the second ancestral population JBB-32 K (clade 4c) probably

migrated from Hokkaido by crossing the Tsugaru Strait at approximately 160 Ka or shortly thereafter (equivalent in age to MIS 6). Subsequently, the eastern and central Hokkaido populations independently migrated from the continent [6].

Although the currently available data is consistent with the Sakhalin-Hokkaido route hypothesis, the Korean route cannot be completely ruled out because there are no population genetic data for the brown bear on the Korean Peninsula or for the Pleistocene individuals in this area. It is still possible that the second ancestral population that diverged from the unknown/extinct population in the continent migrated via the Peninsula, and then expanded northwards across the Tsugaru Strait towards Hokkaido. Also, the possibility that the Honshu and southern Hokkaido brown bear populations migrated to the islands independently from the continent via the Korean and Sakhalin routes, respectively, cannot be completely ruled out. Therefore, a detailed analysis of ancient and modern DNA from Pleistocene brown bear fossils in East Asia will be necessary in the future.

## 4.3. Waves of large mammal migration and the impact on faunal changes in the isolated Japanese Archipelago

Our results also provide insight into the palaeobiogeography of extant brown bears in Hokkaido. The tMRCA values for the central, eastern and southern Hokkaido lineages are estimated as 17 Ka, 26 Ka and 22 Ka, respectively (figure 2). Although they are considered to have migrated to Hokkaido before these times, it remains unclear whether brown bears inhabited Hokkaido throughout the late Pleistocene. This is because almost no terrestrial sediments containing Pleistocene mammal fossils—except some herbivores—have been found in Hokkaido, and therefore there is little palaeontological information about the origin and migration history of Pleistocene mammals in Hokkaido. Our present results suggest that brown bears were present in Hokkaido during, and perhaps before, the early late Pleistocene.

Our results also establish that the stable isotope compositions ($\delta^{13}$C and $\delta^{15}$N) of the Honshu brown bear were very similar to those of the extinct North American short-faced bear from the late Pleistocene in terms of the high proportion of $\delta^{15}$N. Large carnivores that feed on large herbivorous mammals tend to have relatively high $\delta^{15}$N values, as this value tends to be higher for larger mammals in the higher trophic positions (e.g. Rey-Iglesia *et al.* [37]). This result suggests that the Honshu brown bears of the late Pleistocene glacial period were highly carnivorous and highly dependent on large herbivorous mammals, suggesting that they occupied an important trophic position in the middle and late Pleistocene ecosystem. Therefore, the repeated invasion and potential extinction of brown bear populations on Honshu Island would have had an enormous mutual impact also on the reduction and/or extinction of megafauna. Thus, it is possible that faunal exchanges of large mammals between Honshu and the Eurasian continent might have, in general, occurred more frequently than previously thought during the ice ages, which might have shuffled and rearranged the unique ecosystem in the isolated region of the Japanese Archipelago. This is a new, previously inconceivable evolutionary scenario discovered from ancient DNA.

The phylogenetic position of the initially migrating Honshu brown bear, based on the oldest brown bear fossil (approx. 340 Ka), remains uncertain. Because this age predates the tMRCA of clade 4 (203 Ka; 95% HPD: 259–150 Ka) and may be older than that of the eastern clade comprising clades 3, 4 and 5 (288 Ka; 95% HPD: 367–216 Ka), this Chibanian brown bear in Honshu might be one of the earliest diverging lineages among the known populations in the world. It is also unresolved when this ancient population became extinct in Honshu. The second wave of the migration of clade 4c brown bears (approx. 140 Ka) from Hokkaido might have caused population turnover, i.e. affecting the survival of the pre-existing population. It is also notable that the second lineage (clade 4c) again became extinct along with large-bodied herbivores on Honshu Island, such as bison (*Bison priscus*), aurochs (*Bos primigenius*), Naumann's elephant, and Yabe's giant deer before the end of the late Pleistocene.

On the other hand, another ursid species in Honshu, Asiatic black bears (*Ursus thibetanus*), survived the megafaunal extinction event during the late Pleistocene [60]. Palaeontological data suggest they had coexisted with Honshu brown bears from the middle Pleistocene. Because there is almost no overlap of the distribution found for these two ursid species other than at the eastern end of the Eurasian continent (Maritime Territory of Russia), 'sympatric' cooccurrence of these competitive species is one of the anomalous features of Pleistocene megafauna in Japan. In contrast with Honshu brown bears that adapted to carnivory, the Japanese subspecies of Asiatic black bears (*U. t. japonicus*) are more herbivorous, unlike certain other subspecies of Asiatic black bears [61]. Although the ecological

relationship between them—especially during the Pleistocene of Honshu Island—is not well known [1], it is possible that the contrasting dietary dependences seen for these two ursid species were a local adaptation of habit segregation for avoiding competition between them.

Detailed demographic analyses and stable isotope measurements of the Pleistocene mammals in Honshu are prerequisite for elucidating their ecological relationships and for determining why the large-bodied herbivorous and carnivorous mammals became extinct, with the exception of only three medium-bodied mammals, namely the Sika deer (*Cervus nippon*), Japanese serow (*Capricornis crispus*) and Asiatic black bears. Future research, combining palaeontology and ancient DNA analysis including nuclear genome analysis, of the history of mammal migration and extinction will not only elucidate the aforementioned issues but also help us understand how, in general, environmental changes such as sea-level fluctuations during the glacial-interglacial periods affected faunal diversity in geographically isolated ecosystems.

Data accessibility. The raw Illumina sequence data are available under DDBJ DRA ID DRA009015 and DRA012418. The accession number for the mitochondrial genome sequence of the Honshu brown bear fossil specimen (JBB-32 K) reported in this paper is DDBJ/ENA/GenBank: LC595633. The data are provided in electronic supplementary material.

Authors' contributions. N.K., A.K. and F.T. provided the samples and conducted the radiocarbon analysis and the stable isotope analyses. A.A., M.E.A., E.W. and T.S. performed DNA extraction and prepared sequencing libraries and/or provided guidance for these tasks. H.M. performed the bioinformatics analysis. T.Y. and H.N. performed the phylogenetic analyses. N.K. and T.Y. conducted the evolutionary time scale analysis. T.Y. conducted the phylogeographic analysis. T.S., N.K., T.Y., H.M. and H.N. wrote the manuscript. All authors reviewed the manuscript.

Competing Interests. The authors declare no competing interests.

Funding. This study was supported by Grants-in-Aid for Scientific Research (nos 18H04139 and 20K20942) from the Japan Society for the Promotion of Science (JSPS), JSPS Core-to-Core Program (JPJSCCA20170005, Wildlife Research Center of Kyoto University) and the National Institute of Polar Research through the General Collaboration Project (no. 2-31). Radiocarbon dating was supported by the Chemical Stratigraphy and Dating Project of the National Museum of Nature and Science.

Acknowledgements. We thank Jesper Stenderup for producing the analysed sequences, Ukyo Kurosawa and staff of the Ueno Village Board of Education, Hiromichi Kitagawa for providing the sample of the Honshu brown bear, and Jiaqi Wu for the phylogeographic analysis. We also thank the late Shigeru Kobayashi for his donation of the Honshu brown bear fossil to the Saitama Museum of Natural History. Computations were partially performed on the National Institute of Genetics supercomputer at the Research Organization of Information and Systems and the supercomputer system of the Institute of Statistical Mathematics.

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
