## [Peer Review File · Royal Society Open Science]

Review History

RSOS-210518.R0 (Original submission)

Review form: Reviewer 1

Is the manuscript scientifically sound in its present form?

Yes

Are the interpretations and conclusions justified by the results?

Yes

Is the language acceptable?

Yes

Do you have any ethical concerns with this paper?

No

Have you any concerns about statistical analyses in this paper?

Yes

Recommendation?

Accept with minor revision (please list in comments)

Comments to the Author(s)

- 1) Lines 114–115 and elsewhere: Please make clear if the radius illustrated in fig. S1 and referenced in table S1 was used in this study. When you eventually make this issue clear, please try to be consistent about the issue throughout the paper. Specifically, if you eventually decide that the radius was not used, please remove it from the figure and the table.
- 2) Line 133: Please make sure if the reference to fig. 2 fits the text here.
- 3) Lines 229 and 235: Please provide rationale for the models used.
- 4) Lines 245 (Tracer), 258 (BayesTraits), 518 (GMT5) and 545 (ETOP01): Shouldn't these softwares be fully referenced in the References section?
- 5) Section "Estimating coalescence times among *U. arctos*": Admittedly, I've never done a tip-dating analysis with BEAST and therefore can't be sure, but I've done other dating analyses with BEAST and feel that the information provided does not enable replication of your results. Just please make sure if all of the data that are necessary for the replication are provided.
- 6) There may be something wrong with me, but I can't see the accession numbers in Table S1.
- 7) Line 350: There is something wrong here.
- 8) Line 469: I would say that it was rather the megafauna extinction that could have had an impact on the bear extinction.
- 9) Lines 572 (Barnes et al.), 573 (Rey-Iglesia et al.) and 594 (Nguyen et al.): These studies should rather be fully referenced somewhere within the supplementary file.

Review form: Reviewer 2

Is the manuscript scientifically sound in its present form?

Yes

Are the interpretations and conclusions justified by the results?

Yes

Is the language acceptable?

Yes

Do you have any ethical concerns with this paper?

No

Have you any concerns about statistical analyses in this paper?

No

Recommendation?

Accept with minor revision (please list in comments)

Comments to the Author(s)

In general I found the paper accessible and straightforward. As someone interested in this system, I was happy to see new attention focused on it. Based on previous reviews, it appears the authors already put significant effort into improving their manuscript. I have some minor comments that I would encourage addressing to make the paper even more broadly relevant. I was surprised that the authors did not include a comparative discussion of the phylogeography of the Ezo and Japanese (Honsu) wolves, given recent genetic work that also points out complex migration pathways with multiple arrivals (see Matsumara et al. 2014, *Molecular Phylogenetics &*

Evolution and Niemann et al 2021, iScience). I would appreciate seeing further integration of the brown bear migration pathways with other hypothesized mammal biogeographic patterns, perhaps in a figure or summary table. This would be of extreme use to outside researchers who may not have access to the Japanese literature, and would also make clear why this DNA sequence is so valuable. This should include the wolves, some large mammals, but also the small mamma fauna that also shows interesting biogeographic patterns. Also, how does this relate to human arrivals?

Minor

Line 123 - change word "senile" - perhaps not ideal word, negative connotation

Review form: Reviewer 3

Is the manuscript scientifically sound in its present form?

Yes

Are the interpretations and conclusions justified by the results?

Yes

Is the language acceptable?

Yes

Do you have any ethical concerns with this paper?

No

Have you any concerns about statistical analyses in this paper?

No

Recommendation?

Accept as is

Comments to the Author(s)

I believe that the authors have followed the reviewers' suggestions to improve the manuscript, or have responded to them appropriately. I think it is a quality work that provides new data and interpretations on the origin and distribution of brown bears. Congratulations.

Review form: Reviewer 4

Is the manuscript scientifically sound in its present form?

Yes

Are the interpretations and conclusions justified by the results?

Yes

Is the language acceptable?

Yes

Do you have any ethical concerns with this paper?

No

Have you any concerns about statistical analyses in this paper?

No

Recommendation?

Accept with minor revision (please list in comments)

Comments to the Author(s)

The authors present the first ancient DNA data from a Honshu brown bear. These data are used to test hypotheses about the relationships and origin of this extinct population. The authors present a compelling case for the colonisation of Honshu by multiple brown bear populations during the Pleistocene. I enjoyed reading this paper and I congratulate the authors on obtaining ancient DNA data from JBB-32K - I now hope to see more studies of extinct megafauna from Honshu in the future. Overall I see no major problems with the author's data, analysis, or interpretation. However, I have identified some minor points - described below - that I think the authors should address before publication. These should all be straightforward and upon revision I would be happy to recommend that this article be accepted.

Reporting/methodology details

Lines 140-142 - The authors report two new radiocarbon ages. I note that comprehensive raw data are provided in the SI, which is excellent. However, one piece of information that is missing is the pretreatment performed by the radiocarbon lab prior to graphitisation (e.g. ultrafiltration of collagen?). As recently highlighted in RSOC (i.e. <https://doi.org/10.1098/rsos.201351>) this is an important but frequently under-reported detail that affects the reliability of radiocarbon ages based on teeth and bones. The cited article (Tokanai et al., 2011) also does not contain this information. Could the authors obtain these details?

Lines 149-151 - There is some ambiguity here about exactly what work was performed in what laboratory. There are two samples, was one processed in each of the two labs?

Line 174 - The authors mention "two of ... nine experiments" but up until this point only two extractions and one negative control have been mentioned. Could the authors please clarify how many extractions and how many libraries were created from each sample? I see that there are nine experiments mentioned in Table S1, but it is unclear how many of these are different libraries from the same (or different) extractions or alternatively just multiple sequencing runs based on one or more libraries.

Lines 207-218 - Were the libraries sequenced in SR or PE mode? And how were duplicate reads identified? Finally, what exactly is meant by manual curation?

Line 226 - Could the authors please clarify what is meant by "manual adjustment"? As someone who works a lot with mtDNA I think I understand, but this may look a little odd and subjective to some readers.

Line 227 - What are the 10 partitions exactly? Three codon positions for H and L strands is only six partitions by my count. Presumably the others are some combination of RNA/D-loop/non-coding but this isn't clear.

IQ-tree, IQ tree, and IQ-TREE are all used variously to refer to the same piece of software. I think IQ-TREE is the correct usage.

Geographical origin

I have no issues with the authors' conclusion that brown bears migrated to Honshu at least twice - once leading to the population represented by the Chibanian specimen, and once leading to the population represented by JBB-32K and JBB-19K. This is well-founded based on the fossil and genetic data and is the authors' most important finding in my view (and quite rightly makes the title).

On the other hand, I am not particularly convinced either way by the evidence for the route of migration. As the authors concede in lines 442-448, it is equally possible - due to a lack of data from the Eurasian mainland - that clade 4c migrated to Honshu and subsequently Hokkaido via the Korean peninsula. I would even go further to say that Honshu and Hokkaido could in fact have been colonised independently via the Korean peninsula and Sakhalin, respectively, from an ancestral clade 4 population in Pleistocene eastern Asia. This third hypothesis does not require brown bears to have crossed the Blakiston Line at all and would imply that the relationship between the Honshu bear and clade 4 bears from Hokkaido just reflects a lack of ancient data from eastern Asia. For this same reason I'm skeptical about the Bayesian range reconstructions; we know that the observed data are heavily skewed and no doubt inclusion of mainland samples would alter the results. It is certainly interesting and worth mentioning that the age of the split between Honshu and Hokkaido bears (160 Ka) is close to dates inferred for colonisation of Honshu by elephants and deer, but I remain unconvinced based on this comparison alone.

All that being said, I would only suggest that the authors soften their claims about the strength of evidence on this point. For example, in the Abstract perhaps "the second migrated via Hokkaido" could be qualified like this: "the second may have migrated via Hokkaido". Ultimately the authors appropriately acknowledge the uncertainty and need for further research in the Discussion, but some care may be warranted to avoid a casual reader getting the wrong impression.

Typos

Line 46 - "fossil records" should be "fossil record"

Line 67 - "occupy" should be "occupying"

Line 82 - "bears extirpated" should be "bears were extirpated"

Line 90 - I think that "why it is" should be "making it"

Line 114 - "sample" should be "samples"

Line 116 - "brown-bear" should not be hyphenated

Line 119 - I think that "of" should be "a"

Table S1 - "petrosus" should be "petrous"; "Mitochondria" should be "Mitochondrial"

Decision letter (RSOS-210518.R0)

Dear Dr Segawa

On behalf of the Editors, we are pleased to inform you that your Manuscript RSOS-210518 "Ancient DNA reveals multiple origins and migration waves of extinct Japanese brown bear lineages" has been accepted for publication in Royal Society Open Science subject to minor revision in accordance with the referees' reports. Please find the referees' comments along with any feedback from the Editors below my signature.

Please submit your revised manuscript and required files (see below) no later than 7 days from today's (ie 28-Jun-2021) date. Note: the ScholarOne system will 'lock' if submission of the revision is attempted 7 or more days after the deadline. If you do not think you will be able to meet this deadline please contact the editorial office immediately.

on behalf of Dr Emily Lindsey (Associate Editor) and Kevin Padian (Subject Editor)
openscience@royalsociety.org

Associate Editor Comments to Author (Dr Emily Lindsey):

Associate Editor: 1

Comments to the Author:

All four reviewers agreed that this is a well-conducted and -presented study and represents a valuable contribution to the literature. They collectively requested relatively minor edits on the manuscript before acceptance for publication, including:

- Comparison with hypothesized phylogeographic pathways of other Japanese mammals, such as wolves, elephants, deer, small mammals, and humans
- Ensure that all phylogenetic analyses and data are properly presented

- Clarify methods for aDNA extractions and pre-treatment for radiocarbon dates
- Give greater consideration to alternative models for brown bear migration into Japan
- Typographical edits as noted by reviewers

See specific notes from individual reviewers for details. Congratulations on an excellent study and I look forward to receiving your revised manuscript soon.

Associate Editor: 2

Comments to the Author:

This was a generally well-reviewed article that was transferred to RSOS as a more apt venue given the restricted nature of the study. Authors have taken critiques from the previous round of review into account, but the manuscript merits additional peer-review given that only one previous reviewer provided substantive recommendations for revision.

Reviewer comments to Author:

Reviewer: 1

Comments to the Author(s)

- 1) Lines 114-115 and elsewhere: Please make clear if the radius illustrated in fig. S1 and referenced in table S1 was used in this study. When you eventually make this issue clear, please try to be consistent about the issue throughout the paper. Specifically, if you eventually decide that the radius was not used, please remove it from the figure and the table.
- 2) Line 133: Please make sure if the reference to fig. 2 fits the text here.
- 3) Lines 229 and 235: Please provide rationale for the models used.
- 4) Lines 245 (Tracer), 258 (BayesTraits), 518 (GMT5) and 545 (ETOP01): Shouldn't these softwares be fully referenced in the References section?
- 5) Section "Estimating coalescence times among *U. arctos*": Admittedly, I've never done a tip-dating analysis with BEAST and therefore can't be sure, but I've done other dating analyses with BEAST and feel that the information provided does not enable replication of your results. Just please make sure if all of the data that are necessary for the replication are provided.
- 6) There may be something wrong with me, but I can't see the accession numbers in Table S1.
- 7) Line 350: There is something wrong here.
- 8) Line 469: I would say that it was rather the megafauna extinction that could have had an impact on the bear extinction.
- 9) Lines 572 (Barnes et al.), 573 (Rey-Iglesia et al.) and 594 (Nguyen et al.): These studies should rather be fully referenced somewhere within the supplementary file.

Reviewer: 2

Comments to the Author(s)

In general I found the paper accessible and straightforward. As someone interested in this system, I was happy to see new attention focused on it. Based on previous reviews, it appears the authors already put significant effort into improving their manuscript. I have some minor comments that I would encourage addressing to make the paper even more broadly relevant. I was surprised that the authors did not include a comparative discussion of the phylogeography of the Ezo and Japanese (Honshu) wolves, given recent genetic work that also points out complex migration pathways with multiple arrivals (see Matsumara et al. 2014, *Molecular Phylogenetics & Evolution* and Niemann et al 2021, *iScience*). I would appreciate seeing further integration of the brown bear migration pathways with other hypothesized mammal biogeographic patterns, perhaps in a figure or summary table. This would be of extreme use to outside researchers who may not have access to the Japanese literature, and would also make clear why this DNA

sequence is so valuable. This should include the wolves, some large mammals, but also the small mamma fauna that also shows interesting biogeographic patterns. Also, how does this relate to human arrivals?

Minor

Line 123 - change word "senile" - perhaps not ideal word, negative connotation

Reviewer: 3

Comments to the Author(s)

I believe that the authors have followed the reviewers' suggestions to improve the manuscript, or have responded to them appropriately. I think it is a quality work that provides new data and interpretations on the origin and distribution of brown bears. Congratulations.

Reviewer: 4

Comments to the Author(s)

The authors present the first ancient DNA data from a Honshu brown bear. These data are used to test hypotheses about the relationships and origin of this extinct population. The authors present a compelling case for the colonisation of Honshu by multiple brown bear populations during the Pleistocene. I enjoyed reading this paper and I congratulate the authors on obtaining ancient DNA data from JBB-32K - I now hope to see more studies of extinct megafauna from Honshu in the future. Overall I see no major problems with the author's data, analysis, or interpretation. However, I have identified some minor points - described below - that I think the authors should address before publication. These should all be straightforward and upon revision I would be happy to recommend that this article be accepted.

Reporting/methodology details

Lines 140-142 - The authors report two new radiocarbon ages. I note that comprehensive raw data are provided in the SI, which is excellent. However, one piece of information that is missing is the pretreatment performed by the radiocarbon lab prior to graphitisation (e.g. ultrafiltration of collagen?). As recently highlighted in RSOC (i.e. <https://doi.org/10.1098/rsos.201351>) this is an important but frequently under-reported detail that affects the reliability of radiocarbon ages based on teeth and bones. The cited article (Tokanai et al., 2011) also does not contain this information. Could the authors obtain these details?

Lines 149-151 - There is some ambiguity here about exactly what work was performed in what laboratory. There are two samples, was one processed in each of the two labs?

Line 174 - The authors mention "two of ... nine experiments" but up until this point only two extractions and one negative control have been mentioned. Could the authors please clarify how many extractions and how many libraries were created from each sample? I see that there are nine experiments mentioned in Table S1, but it is unclear how many of these are different libraries from the same (or different) extractions or alternatively just multiple sequencing runs based on one or more libraries.

Lines 207-218 - Were the libraries sequenced in SR or PE mode? And how were duplicate reads identified? Finally, what exactly is meant by manual curation?

Line 226 - Could the authors please clarify what is meant by "manual adjustment"? As someone who works a lot with mtDNA I think I understand, but this may look a little odd and subjective to some readers.

Line 227 - What are the 10 partitions exactly? Three codon positions for H and L strands is only six partitions by my count. Presumably the others are some combination of RNA/D-loop/non-coding but this isn't clear.

IQ-tree, IQ tree, and IQ-TREE are all used variously to refer to the same piece of software. I think IQ-TREE is the correct usage.

Geographical origin

I have no issues with the authors' conclusion that brown bears migrated to Honshu at least twice - once leading to the population represented by the Chibanian specimen, and once leading to the population represented by JBB-32K and JBB-19K. This is well-founded based on the fossil and genetic data and is the authors' most important finding in my view (and quite rightly makes the title).

On the other hand, I am not particularly convinced either way by the evidence for the route of migration. As the authors concede in lines 442-448, it is equally possible - due to a lack of data from the Eurasian mainland - that clade 4c migrated to Honshu and subsequently Hokkaido via the Korean peninsula. I would even go further to say that Honshu and Hokkaido could in fact have been colonised independently via the Korean peninsula and Sakhalin, respectively, from an ancestral clade 4 population in Pleistocene eastern Asia. This third hypothesis does not require brown bears to have crossed the Blakiston Line at all and would imply that the relationship between the Honshu bear and clade 4 bears from Hokkaido just reflects a lack of ancient data from eastern Asia. For this same reason I'm skeptical about the Bayesian range reconstructions; we know that the observed data are heavily skewed and no doubt inclusion of mainland samples would alter the results. It is certainly interesting and worth mentioning that the age of the split between Honshu and Hokkaido bears (160 Ka) is close to dates inferred for colonisation of Honshu by elephants and deer, but I remain unconvinced based on this comparison alone.

All that being said, I would only suggest that the authors soften their claims about the strength of evidence on this point. For example, in the Abstract perhaps "the second migrated via Hokkaido" could be qualified like this: "the second may have migrated via Hokkaido". Ultimately the authors appropriately acknowledge the uncertainty and need for further research in the Discussion, but some care may be warranted to avoid a casual reader getting the wrong impression.

Typos

Line 46 - "fossil records" should be "fossil record"

Line 67 - "occupy" should be "occupying"

Line 82 - "bears extirpated" should be "bears were extirpated"

Line 90 - I think that "why it is" should be "making it"

Line 114 - "sample" should be "samples"

Line 116 - "brown-bear" should not be hyphenated

Line 119 - I think that "of" should be "a"

Table S1 - "petrosus" should be "petrous"; "Mitochondria" should be "Mitochondrial"

===PREPARING YOUR MANUSCRIPT===

===PREPARING YOUR REVISION IN SCHOLARONE===

-- If you have uploaded ESM files, please ensure you follow the guidance at <https://royalsociety.org/journals/authors/author-guidelines/#supplementary-material> to include a suitable title and informative caption. An example of appropriate titling and captioning may be found at https://figshare.com/articles/Table_S2_from_Is_there_a_trade-off_between_peak_performance_and_performance_breadth_across_temperatures_for_aerobic_scops_in_teleost_fishes_/3843624.

Author's Response to Decision Letter for (RSOS-210518.R0)

See Appendix A.

Decision letter (RSOS-210518.R1)

Dear Dr Segawa,

I am pleased to inform you that your manuscript entitled "Ancient DNA reveals multiple origins and migration waves of extinct Japanese brown bear lineages" is now accepted for publication in Royal Society Open Science.

on behalf of Dr Emily Lindsey (Associate Editor) and Kevin Padian (Subject Editor)
openscience@royalsociety.org

Appendix A

Reviewer comments to Author:

Reviewer: 1

Comments to the Author(s)

1) Lines 114–115 and elsewhere: Please make clear if the radius illustrated in fig. S1 and referenced in table S1 was used in this study. When you eventually make this issue clear, please try to be consistent about the issue throughout the paper. Specifically, if you eventually decide that the radius was not used, please remove it from the figure and the table.

→ Thank you for the comments. We used the radius illustrated in fig. S1 and referenced in table S1 in this study (lines 111-113). We have added a sentence to describe the number of uniquely mapped mitochondrial reads from the petrous and the radius in Result 3.2. We have modified table S1 to clarify both of the bones were used for the reconstruction of the mitochondrial genome sequence. In any case, the radius was also used for C14 analysis to determine the geochronological age of this individual, so the illustration is necessary.

2) Line 133: Please make sure if the reference to fig. 2 fits the text here.

→ Thank you for the comments. We revised “fig. 2” to “electronic supplementary material, figure S1” (line 130).

3) Lines 229 and 235: Please provide rationale for the models used.

→ Thank you for the comments. Indeed, model selections (e.g., nucleotide model vs. codon model, codon-partition model vs. noncodon-partition model, Γ model vs. no- Γ model) are substantially important for the phylogenetic inferences and divergence time estimations among higher taxonomic groups such as Family, Order or Class (e.g., Li et al. 2013, Wu et al. 2014). Since the effect of the multiple substitutions is significant among this level of taxonomic groups, the model fits in better with data should be selected for more accurate inference of the numbers of multiple substitutions. On the other hand, the effect of the multiple substitutions is generally not very serious in polymorphisms within a single species. In such a case, the impact of model selections is limited for the phylogenetic inferences and coalescent time estimations (Yang 2015). In our phylogenetic inference, we applied the GTR+I+ Γ model for all partitions because all other nucleotide substitution models are the special cases of this model. On the other hand, we applied the HKY+I+ Γ model for the coalescent time estimation to avoid the computational burden. Because we need to conduct sufficiently long MCMC for convergence of all parameters, the parameter-rich model like the GTR model needs extremely long MCMC for our data set (16,084 sites with 10 partitions \times 96 individuals). Therefore, we used a simpler model for the better convergence of the parameters. Although the HKY model (4 parameters) is simpler than the GTR model (8 parameters), the HKY model takes account of the nucleotide

composition bias (especially serious in the third codon position of mtgenome) and transition/transversion ratio (Hasegawa et al. 1985). We think HKY model is sufficient for this analysis.

Furthermore, the partition strategies (this is also “model selection”) impact more seriously in terms of the numbers of the parameters. We carefully applied the PartitionFinder program and selected the best partition strategy. In addition, we excluded the hyper-variable regions of D-loop, where the effect of the multiple substitutions are most serious in mtgenome, from the phylogenetic inference and coalescent time estimations.

Y Li, Z Ren, AM Shedlock, J Wu, L Sang, T Tersing, M Hasegawa, T Yonezawa, Y Zhong (2013) High altitude adaptation of the schizothoracine fishes (Cyprinidae) revealed by the mitochondrial genome analyses. *Gene*. 517: 169-178

J Wu, M Hasegawa, Y Zhong, T Yonezawa (2014) Importance of synonymous substitutions under dense taxon sampling and appropriate modeling in reconstructing the mitogenomic tree of Eutheria *Genes & genetic systems* 89: 237-251

Z Yang (2015) The BPP program for species tree estimation and species delimitation. *Current Zoology* 61: 854-865

M Hasegawa, H Kishino, T Yano (1985) Dating of the human-ape splitting by a molecular clock of mitochondrial DNA. *Journal of molecular evolution* 22: 160-174

4) Lines 245 (Tracer), 258 (BayesTraits), 518 (GMT5) and 545 (ETOP01): Shoudn't these softwares be fully referenced in the References section?

→ Thank you for the comment. We described the URL for the software in the manuscript because they are not published as a paper but available online only.

5) Section “Estimating coalescence times among *U. arctos*”: Admittedly, I've never done a tip-dating analysis with BEAST and therefore can't be sure, but I've done other dating analyses with BEAST and feel that the information provided does not enable replication of your results. Just please make sure if all of the data that are necessary for the replication are provided.

→ Thank you for the comments. Because the C14 dates of ancient specimen themselves work as time-calibrations for coalescent time estimations, we did not provide any other prior information. Therefore, current information is sufficient for repeating our analysis.

6) There may be something wrong with me, but I can't see the accession numbers in Table S1.

→ Thank you for the comments. We have added the accession number to Table S1.

7) Line 350: There is something wrong here.

→ Thank you for the comments. We rewrote it (line 359).

8) Line 469: I would say that it was rather the megafauna extinction that could have had an impact on the bear extinction.

→ Thank you for the comments. We also considered the impact of the megafaunal extinction and/or reduction for the extinction of brown bears on Honshu Island. However, there is, at the moment, no sufficient evidence of repeated, "simultaneous" extinctions of megafauna on Honshu Island during the middle and late Pleistocene, or the timings of the extinctions of megafauna and brown bears did not show any geochronological synchronisities based on available fossil records (for instance, 18,000 BP as the youngest record of brown bears but 12,000 BP as the youngest record of megafauna) on Honshu Island. However, the potential impact of a reduction and/or extinction of megafauna to an extinction of brown bears will not be able to ignore as was pointed out from the reviewer. In this regard, we slightly modified the discussion as "Therefore the repeated invasion and potential extinction of brown bear populations on Honshu Island would have had an enormous mutual impact also on the reduction and/or extinction of megafauna" (lines 477-479) .

9) Lines 572 (Barnes et al.), 573 (Rey-Iglesia et al.) and 594 (Nguyen et al.): These studies should rather be fully referenced somewhere within the supplementary file.

→ There is a supplementary references section at the end of the supplement file, and we have cited them there.

Reviewer: 2

Comments to the Author(s)

In general I found the paper accessible and straightforward. As someone interested in this system, I was happy to see new attention focused on it. Based on previous reviews, it appears the authors already put significant effort into improving their manuscript. I have some minor comments that I would encourage addressing to make the paper even more broadly relevant.

I was surprised that the authors did not include a comparative discussion of the phylogeography of the Ezo and Japanese (Honshu) wolves, given recent genetic work that also points out complex migration pathways with multiple arrivals (see Matsumara et al. 2014, *Molecular Phylogenetics & Evolution* and Niemann et al 2021, *iScience*). I would appreciate seeing further integration of the brown bear migration pathways with other hypothesized mammal biogeographic patterns, perhaps in a figure or summary table. This would be of extreme use to outside researchers who may not have access to the Japanese literature, and would also make clear why this DNA sequence is so valuable. This should include the wolves, some large mammals, but also the small mamma fauna that also shows interesting biogeographic patterns. Also, how does this relate to human arrivals?

→ Thank you for the comments. As you pointed out, the route by which Pleistocene mammals migrated from the Eurasian continent to the Japanese archipelago is an important question. It is clear that some large mammals, such as the Hokkaido brown bear populations (clades 3a2 and 3b) and Ezo wolf migrated via the Sakhalin route. However, for the mammals inhabited in Honshu, even the Japanese wolf, there is no published evidence regarding the migration route even in the Japanese papers. Therefore, to avoid the readers' confusion, we decided not to discuss the migration route of other mammals in this paper. We also hope that a combined study of paleontology and paleogenetics will elucidate the entire migration history of the Pleistocene mammals along with the paleogeography of the Japanese Archipelago.

Minor

Line 123 - change word "senile" - perhaps not ideal word, negative connotation

→ Thank you for the comment. We reworded this as "old adult female" (line 120) .

Reviewer: 3

Comments to the Author(s)

I believe that the authors have followed the reviewers' suggestions to improve the manuscript, or have responded to them appropriately. I think it is a quality work that provides new data and interpretations on the origin and distribution of brown bears. Congratulations.

→ Thank you very much for reviewing our manuscript.

Reviewer: 4

Comments to the Author(s)

The authors present the first ancient DNA data from a Honshu brown bear. These data are used to test hypotheses about the relationships and origin of this extinct population. The authors present a compelling case for the colonisation of Honshu by multiple brown bear populations during the Pleistocene. I enjoyed reading this paper and I congratulate the authors on obtaining ancient DNA data from JBB-32K - I now hope to see more studies of extinct megafauna from Honshu in the future. Overall I see no major problems with the author's data, analysis, or interpretation. However, I have identified some minor points - described below - that I think the authors should address before publication. These should all be straightforward and upon revision I would be happy to recommend that this article be accepted.

→ Thank you for the positive comments. We are glad to know that you enjoyed this work. We addressed your comments as below.

Reporting/methodology details

Lines 140-142 - The authors report two new radiocarbon ages. I note that comprehensive raw data are provided in the SI, which is excellent. However, one piece of information that is missing is the pretreatment performed by the radiocarbon lab prior to graphitisation (e.g. ultrafiltration of collagen?). As recently highlighted in RSOC (i.e. <https://doi.org/10.1098/rsos.201351>) this is an important but frequently under-reported detail that affects the reliability of radiocarbon ages based on teeth and bones. The cited article (Tokanai et al., 2011) also does not contain this information. Could the authors obtain these details?

→ Thank you for the comment. In the main text, we added the detail of pretreatment protocol to extract and purify the gelatinized collagen from bone and tooth samples (at the National Museum of Nature and Science) to guarantee the reliability of the pretreatment itself (lines 135-145).

Lines 149-151 - There is some ambiguity here about exactly what work was performed in what laboratory. There are two samples, was one processed in each of the two labs?

→ We thank the reviewer's comments. We have revised our manuscript according to the reviewer's comments (lines 155-157).

Line 174 - The authors mention "two of ... nine experiments" but up until this point only two extractions and one negative control have been mentioned. Could the authors please clarify how many extractions and how many libraries were created from each sample? I see that there are nine experiments mentioned in Table S1, but it is unclear how many of these are different libraries from the same (or different) extractions or alternatively just multiple sequencing runs based on one or more libraries.

→ We thank the reviewer for this comment. We rewrote as follows; "from two (Experiment 7 and Experiment 8 in electronic supplementary material, table S1) of the eleven experiments of extracted DNA" (lines 179-180) and "For the remaining nine experiments, the genome libraries were generated without removing the most frequent post-mortem DNA damage" (lines 189-190). Also, to avoid confusion, the Experiment No. column in TableS1 has been modified.

Lines 207-218 - Were the libraries sequenced in SR or PE mode? And how were duplicate reads identified? Finally, what exactly is meant by manual curation?

→ Thank you for the comments. We conducted the single-read sequencing analysis in this study. The PCR duplicate reads (the reads that have a completely identical sequence) were removed using the rmdup function of seqkit. The manual curation in this study means that we manually replaced the nucleotides to N in the site that the read coverage ≤ 0 . We have added the sentences to describe these methods in Material and methods 2.4 (lines 214-224).

Line 226 - Could the authors please clarify what is meant by "manual adjustment"? As someone who works a lot with mtDNA I think I understand, but this may look a little odd and subjective to some readers.

→ Thank you for the comments. We rephrased the sentence as follows: ", and misaligned segments of the alignment were then corrected." (line 232).

Line 227 - What are the 10 partitions exactly? Three codon positions for H and L strands is only six partitions by my count. Presumably the others are some combination of RNA/D-loop/non-coding but this isn't clear.

→ We apologize for the lack of detailed explanation. You are right. Three codon positions for H and L strands is six partitions. In addition, H-strand tRNAs, L-strand tRNAs, rRNAs, and the D-loop region should be counted. PartitionFinder2 agreed with this partitioning. We added the information in the manuscript as follows: "Sequence data for 16,084 sites were separated into 10 partitions according to the three codon positions and the mtDNA coding strand, as well as tRNAs (H- and L- strand), rRNAs, and the D-loop region, in accordance with the PartitionFinder2 [32]." (lines 232-235).

IQ-tree, IQ tree, and IQ-TREE are all used variously to refer to the same piece of software. I think IQ-TREE is the correct usage.

→ Thank you for the comment. We reworded them as 'IQ-TREE' throughout the manuscript.

Geographical origin

I have no issues with the authors' conclusion that brown bears migrated to Honshu at least twice - once leading to the population represented by the Chibanian specimen, and once leading to the population represented by JBB-32K and JBB-19K. This is well-founded based on the fossil and genetic data and is the authors' most important finding in my view (and quite rightly makes the title).

On the other hand, I am not particularly convinced either way by the evidence for the route of migration. As the authors concede in lines 442-448, it is equally possible - due to a lack of data from the Eurasian mainland - that clade 4c migrated to Honshu and subsequently Hokkaido via the Korean peninsula. I would even go further to say that Honshu and Hokkaido could in fact have been colonised independently via the Korean peninsula and Sakhalin, respectively, from an ancestral clade 4 population in Pleistocene eastern Asia. This third hypothesis does not require brown bears to have crossed the Blakiston Line at all and would imply that the relationship between the Honshu bear and clade 4 bears from Hokkaido just reflects a lack of ancient data from eastern Asia. For this same reason I'm skeptical about the Bayesian range reconstructions; we know that the observed data are heavily skewed and no doubt inclusion of mainland samples would alter the results. It is certainly interesting and worth mentioning that the age of the split between Honshu and Hokkaido bears (160 Ka) is close to dates inferred for colonisation of Honshu by elephants and deer, but I remain unconvinced based on this comparison alone.

→ Thank you for valuable comments.

As seen in Figure S7, we retrieved all available *cytochrome b* data and D-loop data of *U. arctos* and conducted the phylogenetic analysis. We found none of the Eurasian mainland samples are included within clade 4. Accordingly, the samples used for Bayesian timetree (Figure 2) well represent the geographical distributions of clade 4. Therefore, we suggest the Sakhalin route as the best implication as inferred from the current distribution patters of *U. arctos* and their genealogical relationships. However, since the Korean route and the independent routes that you kindly suggested cannot be excluded from our current data, we discussed about all the possibility in the revised Discussion part (lines 450-458).

All that being said, I would only suggest that the authors soften their claims about the strength of evidence on this point. For example, in the Abstract perhaps "the second migrated via Hokkaido" could be qualified like this: "the second may have migrated via Hokkaido". Ultimately the authors appropriately acknowledge the uncertainty and need for further research in the Discussion, but some care may be warranted to avoid a casual reader getting the wrong impression.

→ We thank the reviewer's constructive comments. According to the reviewer's suggestion, we mentioned about the third hypothesis for migration, in which brown bears on Honshu and Hokkaido might have migrated independently from the Eurasian continent through the Korean route and Sakhalin route, respectively (lines 455-457). Therefore, by showing all the three possible hypotheses in the manuscript, the Sakhalin route has been relatively weakened in the revised Discussion part. As discussed in the manuscript, however, the Sakhalin route is still more favorable than the other hypotheses by our analyses, based on the results of BayesTrait, the divergence time that corresponds to the time when other mammals crossed the Blakiston Line, and the fact that the other Hokkaido brown bears (the Central and Eastern populations) migrated via the Sakhalin route. Therefore, we left the phrase in the abstract as it was.

Typos

Line 46 - "fossil records" should be "fossil record"

We reworded it to "fossil record" (line 45).

Line 67 - "occupy" should be "occupying"

→ We think "occupy" is appropriate in this sentence (line 65).

Line 82 - "bears extirpated" should be "bears were extirpated"

→ We rephrased it to "bears were extirpated" (line 80).

Line 90 - I think that "why it is" should be "making it"

→ We rephrased it to "making it" (line 88).

Line 114 - "sample" should be "samples"

→ We corrected it to "samples" (line 111).

Line 116 - "brown-bear" should not be hyphenated

→ We rephrased it to "brown bear" (line 113).

Line 119 - I think that "of" should be "a"

→ We reworded it to "a" (line 116).

Table S1 - "petrosus" should be "petrous"; "Mitochondria" should be "Mitochondrial"

→ We revised "petrosus" to "petrous" throughout the manuscript. Also, "Mitochondria" was corrected to "Mitochondrial".

Thank you for all the valuable suggestions.